# Learning to Balance Altruism and Self-interest Based on Empathy in Mixed-Motive Games

**Fanqi Kong**[2,1]  **Yizhe Huang**[2,1]  **Song-Chun Zhu**[1,2,3]  **Siyuan Qi**[1]  **Xue Feng** ✉[1]

[1]State Key Laboratory of General Artificial Intelligence, BIGAI
[2]Institute for Artificial Intelligence, Peking University
[3]Department of Automation, Tsinghua University
`kfq20@stu.pku.edu.cn, fengxue@bigai.ai`

## Abstract

Real-world multi-agent scenarios often involve mixed motives, demanding altruistic agents capable of self-protection against potential exploitation. However, existing approaches often struggle to achieve both objectives. In this paper, based on that empathic responses are modulated by inferred social relationships between agents, we propose LASE (**L**earning to balance **A**ltruism and **S**elf-interest based on **E**mpathy), a distributed multi-agent reinforcement learning algorithm that fosters altruistic cooperation through gifting while avoiding exploitation by other agents in mixed-motive games. LASE allocates a portion of its rewards to co-players as gifts, with this allocation adapting dynamically based on the social relationship — a metric evaluating the friendliness of co-players estimated by counterfactual reasoning. In particular, social relationship measures each co-player by comparing the estimated $Q$-function of current joint action to a counterfactual baseline which marginalizes the co-player's action, with its action distribution inferred by a perspective-taking module. Comprehensive experiments are performed in spatially and temporally extended mixed-motive games, demonstrating LASE's ability to promote group collaboration without compromising fairness and its capacity to adapt policies to various types of interactive co-players.

## 1 Introduction

Multi-agent reinforcement learning (MARL) has exhibited impressive performance in numerous collaborative tasks and zero-sum games such as MPE, StarCraft, and Google Research Football [19, 25, 38]. These environments involve a predefined competitive or cooperative relationship between agents. Besides, mixed-motive games are prevalent, in which the relationships between agents are non-deterministic and dynamic. That is, agents could cooperate with some co-players and simultaneously compete with someone else. Furthermore, along with interactions, friends may turn into foes, and vice versa. In such games, to maximize self-interest, agents need to cooperate altruistically in some relationships while keep self-interested to avoid being exploited in some others. Consequently, in mixed-motive environments, the ability to balance altruism and self-interest according to social relationships is crucial for agent performance.

The commonly used CTDE (Centralized Training and Decentralized Execution) methods [31, 29] in MARL focus on global optimization goals and necessitate individual information sharing with centralized controllers, which is impractical for self-interest agents in mixed-motive games. On the other hand, simply training self-interest agents in a decentralized way may converge to local

---

✉Corresponding author.

38th Conference on Neural Information Processing Systems (NeurIPS 2024).

optima, failing to maximize individual interests. For example, in Iterated Prisoner's Dilemma (IPD), decentralized A2C agents converge to defection, getting the minimal reward 0 (see details in Fig. 4).

Within the framework of decentralized learning, the gifting mechanism has been used to address the decision-making problems in mixed-motive games by enabling agents to transfer a portion of their rewards to others [20, 4]. Agents independently select gift recipients and determine gift amount, which complies with the decentralized requirement. Gifting can potentially shape co-players' policies and even incentivize them to behave more altruistically by influencing co-players' reward structure. Previous work has studied handcrafted gifting scheme [20, 34] and has learned end-to-end neural networks to determine the reward transfer scheme [36, 37]. However, a consideration of the correlation between social relationships and response strategies is absent, which is crucial for decision-making in mixed-motive games.

To balance the dilemma between altruism and self-interest caused by the non-deterministic and dynamic relationship between agents, it is a feasible way that adaptively modulating the gift amount to others according to the social relationships. This process is called (cognitive) empathy in developmental psychology[28]. What's more, previous studies and human behavioral experiments have shown that empathy can promote the emergence of altruism among self-interested individuals [1, 2, 30]. To the best of our knowledge, there has been a lack of computational models of empathy and the study of empathy-based decision-making. In this work, we propose a computational model of empathy, in which social relationship is measured by a continuous variable, capturing the influence of co-players' behavior on the focal agent's reward and guiding gift scheme. On that basis, we provide a distributed MARL algorithm LASE (**L**earning to balance **A**ltruism and **S**elf-interest based on **E**mpathy) to address the dilemma between altruism and self-interest in mixed-motive games.

LASE uses counterfactual reasoning to infer the different social relationships with different co-players separately by comparing the estimated $Q$-value for the joint action to a counterfactual baseline established for each other agent. The counterfactual baseline marginalizes a single agent's action while keeping the other agents' actions fixed. This computational approach to social relationships enables LASE to explicitly decompose the value contributions of other agents to it, thereby providing clearer guidance for gift allocation. That is, gift more to the co-players who contribute more. Additionally, to deal with the challenge of inferring co-players in partially observable and decentralized environments, LASE is equipped with a perspectivec taking module to predict others' policies by converting LASE's local observation to a simulated observation of others.

To verify the effectiveness of LASE, we theoretically analyze its dynamics of decision-making in iterated mixed-motive games and conduct comprehensive experiments in spatially and temporally extended mixed-motive games. The results demonstrate LASE's ability to promote group cooperation without compromising fairness and its capability of self-protection against potential exploitation.

This paper makes three main contributions. **(1)** To our best knowledge, we are the first to computationally model empathy which modulates the response based on inferred social relationships. **(2)** We present LASE, a decentralized MARL algorithm that balances altruism and self-interest in mixed-motive games. It flexibly adapts strategy to promote cooperation while mitigating exploitation by others. **(3)** We provide a theoretical analysis of decision dynamics in iterated matrix games and experimentally verify that LASE outperforms baselines in a variety of sequential social dilemmas.

## 2 Related work

In multi-agent learning, the game-theoretic notion of social dilemmas has been generalized from the classic two-player matrix-form games Tab. 1 to sequential social dilemmas, a spatial-temporally-extended complex behavior learning setting [15, 4]. Various approaches have been proposed to foster cooperative behavior among agents to advance societal welfare. One approach incorporates the rewards of others as intrinsic rewards into one's own optimization objectives, with the ratio of intrinsic to extrinsic rewards determined by different methods, such as pre-defined social value orientations like altruistic or prosocial [21, 23], introducing the concept of inequity aversion [11], or learning in a model-free way [33]. However, this approach depends on direct access to others' reward functions, which may not be feasible in realistic mixed-motive games. Another line of work dispenses with this assumption, allowing agents to model others and influence them through their actions [6, 17, 13, 8, 10]. Here, we employ a more direct form of opponent shaping named gifting.

As a peer rewarding mechanism that allows agents to reward other agents as a part of their action space [20], gifting can be viewed as a process of redistributing rewards among agents [12, 9, 7], but the key difference is that in our work, gifting does not require a powerful centralized controller to decide on the allocation. Instead, individuals make their own decisions about gifting, which is more in line with the setting of decentralized training. [34, 35] study the theoretical basis of how gifting promotes cooperative behavior in simple social dilemmas, while LIO [36] independently learns an incentive function to gift others. However, the rewards used for gifting in LIO are determined by the incentive function rather than split from its own reward. LIO doesn't adhere to zero-sum conditions, which to some extent alters the original game-theoretic nature. LToS [37] models the optimization problem of gifting (sharing) weights with the zero-sum setting as a bi-level problem and uses an end-to-end approach to train weights and policies jointly. MOCA [3] introduces contracts to restrict gifting recipients to the agents that fulfill certain behavioral patterns.

Our perspective taking module simulates others' behavior by adopting their perspective, a technique akin to Self-Other Modeling (SOM) [24]. The difference is that instead of inferring the agent's goal which may not necessarily be well defined in some environments, we directly imagine their observations, and decouple the agent's network for inferring others' actions from its own real policy network utilized for execution. In MARL, some prior studies have employed counterfactual reasoning to deduce the impact of individual actions on others [13] or the whole team [5]. In contrast, our focus lies on assessing others' influence on the focal agent.

## 3 Preliminaries

### 3.1 Partially observable Markov games

We consider an $N$-player partially observable Markov game (POMG) [27, 18], $\mathcal{M} = \langle N, \mathcal{S}, \{\mathcal{O}^i\}, \{\mathcal{A}^i\}, \mathcal{P}, \{\mathcal{R}^i\} \rangle$, where $N$ represents the number of agents, $s \in \mathcal{S}$ represents the state of the environment. In the partially observable setting, agent $i$ only obtains the local observation $o^i \in \mathcal{O}^i$ based on the current state $s$. Each agent $i$ learns an independent policy $\pi^i(a^i|o^i)$ to select actions and form a joint action $\boldsymbol{a} = (a^1, ..., a^N) \in \mathcal{A}^1 \times \cdots \times \mathcal{A}^N$, resulting in the state change from $s$ to $s'$ according to the transition function $\mathcal{P} : \mathcal{S} \times \mathcal{A}^1 \times \cdots \times \mathcal{A}^N \to \Delta(\mathcal{S})$, where $\Delta(\mathcal{S})$ represents a probability distribution over the set $\mathcal{S}$. Agent $i$ receives an individual extrinsic reward $r^i = \mathcal{R}^i(s, a^1, \cdots, a^N)$ and tries to maximize a long-term return:

$$V_i^{\boldsymbol{\pi}}(s_0) = \mathbb{E}_{\boldsymbol{a_t} \sim \boldsymbol{\pi}, s_{t+1} \sim P(s_t, \boldsymbol{a_t})} \left[ \sum\nolimits_{t=0}^{\infty} \gamma^t \mathcal{R}^i(s_t, \boldsymbol{a_t}) \right], \tag{1}$$

where the variables in bold represent the joint information of all agents, and $\gamma$ is the discount factor.

### 3.2 Policy Gradient Learning

In decentralized MARL, each self-interest agent $i$ learns an independent policy $\pi^i$ parameterized by $\theta^i$. The optimization objective is to maximize the expected return in Eq. 1. We use the policy-based Actor-Critic method as the learning algorithm for our agents. The gradient for actor is $\nabla_\theta J(\theta) = \mathbb{E}_{\pi_\theta}[\sum_{t=0}^{T} \psi_t \nabla_\theta \log \pi_\theta(a_t|o_t)]$, where $\psi_t$ represents the critic's evaluation of the actor. The most popular form of $\psi_t$ is TD-error: $\psi_t = r_t + \gamma V^{\pi_\theta}(s_{t+1}) - V^{\pi_\theta}(s_t)$.

## 4 Methodology

To balance altruism and self-interest in mixed-motive games, we propose a distributed MARL algorithm LASE which empathically shares rewards with co-players based on inferred social relationships. The architecture of LASE is illustrated in Fig. 1, composed of two main modules: Social Relationship Inference (SRI) and Gifting. SRI conducts counterfactual reasoning to get the social relationships with co-players, which reflects the impact of co-players' actions on LASE's return. SRI compares the $Q$-value (estimated by the SR value network) of the current joint action to a counterfactual baseline which marginalizes the co-player's action, with its action distribution inferred by a perspective-taking (PT) module. PT is provided to address the challenge of predicting co-players' policies in partially observable and decentralized environments. In particular, PT consists of an observation conversion network, simulating the co-player $j$'s observation $\hat{o}^j$ from the local observation $o^i$, and an SR policy network, learning a function from $\hat{o}^j$ to the inferred $j$'s policy.

The Gifting module, according to the inferred social relationships, determines the amount of reward sharing with others. Meanwhile, agents receive others' gifts and get the final reward $r^{i,\text{tot}}$, which

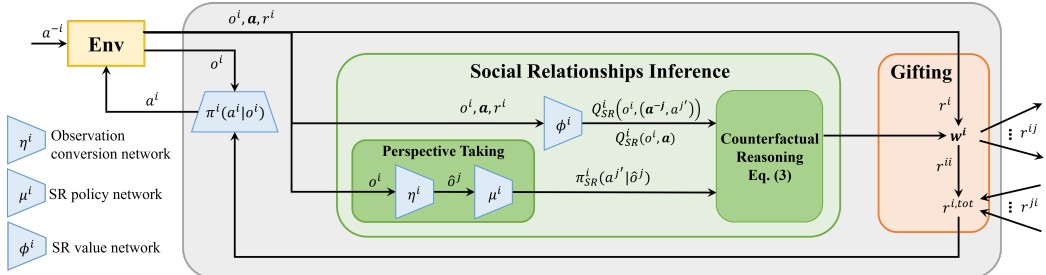

Figure 1: Architecture of LASE. It consists of Social Relationships Inference (SRI) and Gifting. SRI conducts counterfactual reasoning to get the social relationships with co-players. The social relationship is measured by comparing the $Q$-value (estimated by the SR value network) of the current joint action to a counterfactual baseline which marginalizes the co-player's action, with its action distribution inferred by a perspective-taking (PT) module. PT is provided to address the challenge of predicting co-players' policies in partially observable and decentralized environments. The Gifting module, according to the inferred social relationships, determines the amount of reward to share.

serves as an optimization target to guide the training of policy $\pi^i$. Section 4.1 and Section 4.2 will introduce our algorithm in detail. LASE's pseudocode is given as Algorithm 1.

## 4.1 Zero-Sum Gifting

Here, we use the zero-sum gifting mechanism which adheres to the principle of "what I give is what I lose" [20], indicating that the overall rewards for the group remain constant. Notably, gifting is an autonomous decision, wherein any agent $i$ holds a gifting weight vector at time step $t$: $\boldsymbol{w_t^i} = [w_t^{ij}]_{j=1}^N$, where $w_t^{ij} \in [0,1]$ and $\sum_{j=1}^N w_t^{ij} = 1$. $w_t^{ij}$ is the fraction of agent $i$'s reward to gift $j$. It is exactly the social relationship computed as Eq. 3. For an $N$-player group using the zero-sum gifting mechanism, $\boldsymbol{r_t}$ denotes the extrinsic rewards vector obtained through interactions with the environment at timestep $t$. Agent $i$'s total reward is computed by

$$r_t^{i,\text{tot}}(\boldsymbol{w_t}, \boldsymbol{r_t}) = \sum_{j=1}^N w_t^{ji} r_t^j. \tag{2}$$

The policy $\pi^i(a_t^i|o_t^i)$ is trained to maxmize $\mathbb{E}_{\pi^i}[\sum_{t=0}^T \gamma^t r_t^{i,\text{tot}}]$ using the TD-error introduced in Section 3.2 by replacing $r_t$ with $r_t^{\text{tot}}$.

## 4.2 Social Relationships Inference

The social relationship $w^{ij}$, modeled as a continuous variable, measures $i$'s inference of $j$'s friendliness to him. Based on counterfactual reasoning, $w^{ij}$ is inferred as:

$$w^{ij} = \frac{Q_{\text{SR}}^i(o_t^i, \boldsymbol{a_t}) - \sum_{a_t^{j'}} \pi_{\text{SR}}^i(a_t^{j'}|\hat{o}_t^j) Q_{\text{SR}}^i(o_t^i, (\boldsymbol{a_t^{-j}}, a_t^{j'}))}{\mathcal{M}}, \tag{3}$$

where $Q_{\text{SR}}^i(o_t^i, \boldsymbol{a_t})$, estimated by the SR value network, is $i$'s $Q$-value of its local observation and the joint action. Eq. 3 compares $Q_{\text{SR}}^i(o_t^i, \boldsymbol{a_t})$ with a counterfactual baseline, which is the weighted sum of $Q$-values, with $j$ taking all possible actions $a_t^{j'}$ while the other agents' actions $\boldsymbol{a}^{-j}$ fixed. The weight $\pi_{\text{SR}}^i(a_t^{j'}|\hat{o}_t^j)$ is $j$'s policy inferred by $i$. The denominator is for normalization, $\mathcal{M} = (N-1)(\max_{a_t^{j'}} Q_{\text{SR}}^i(o_t^i, (\boldsymbol{a_t^{-j}}, a_t^{j'})) - \min_{a_t^{j'}} Q_{\text{SR}}^i(o_t^i, (\boldsymbol{a_t^{-j}}, a_t^{j'})))$, where $(N-1)$ ensures $w^{ij} \leq \frac{1}{N-1}$. After gifting, LASE keeps the remaining reward for itself, $w^{ii} = 1 - \sum_{j=1, j \neq i}^N w^{ij}$.

Due to partial observability and decentralized learning, $i$ is unable to accurately obtain $j$'s policy $\pi^j$ and its observation $o_t^j$. So we utilize PT module to estimate $j$'s policy, denoted as $\pi_{\text{SR}}^i(a_t^{j'}|\hat{o}_t^j)$. $\pi_{\text{SR}}^i$ is the SR policy network parameterized by $\mu^i$ which predicts $j$'s actions conditioned on the simulated obervation $\hat{o}_t^j$ learned by the observation conversion network. The observation conversion network, parameterized by $\eta^i$, enables LASE to adopt the perspective of others and generate a simulated observation of them. At timestep $t$, LASE processes its own observation $o_t^i$ along with another agent

$j$'s ID (represented as a one-hot vector), yielding the output $\hat{o}_t^j$ which is of the same size as $o_t^i$. To update $\eta^i$ and get a more accurate $\hat{o}_t^j$, the loss function is

$$\mathcal{L}(\eta^i) = \sum_t \sum_{j=1, j \neq i}^N (1-\delta) CE(\mathbb{I}\{a_t^j\}, \pi_{\text{SR}}^i(a_t^j|\hat{o}_t^j)) + \delta\|\hat{o}_t^j - o_t^i\|_1. \qquad (4)$$

The first term aims to ensure that the predicted policy $\pi_{\text{SR}}^i(a_t^j|\hat{o}_t^j)$ aligns with co-player $j$'s actual actions. The second term aims to minimize the deviation of the simulated observation $\hat{o}_t^j$ from $i$'s true observation $o_t^i$, so that some common features in the environment can be reconstructed. The hyperparameter $\delta \in [0, 1]$ balances the two goals.

Since SR policy network predicts actions from an agent's self-perspective and SR value network estimates the agent's own returns, we integrated the training processes of the two networks within an actor-critic framework. In training, $\pi_{\text{SR}}^i$ takes one agent's observation as input and outputs a probability distribution over his action space. $Q_{\text{SR}}^i$ computes the $Q$-value of the joint action under the current observation. Both networks are updated based on the individual extrinsic rewards obtained in the environment, and the TD-error defined as $\delta_t^i = r_t^i + \gamma Q_{\text{SR}}^i(o_{t+1}^i, \boldsymbol{a_{t+1}}) - Q_{\text{SR}}^i(o_t^i, \boldsymbol{a_t})$. Specifically, the SR policy network and SR value network, parameterized by $\mu^i$ and $\phi^i$, are updated by

$$\mu^i = \mu^i + \alpha_{\mu^i} \sum_t \delta_t^i \nabla_{\mu^i} \log \pi_{\text{SR}}^i(a_t^i|o_t^i), \quad \phi^i = \phi^i + \alpha_{\phi^i} \sum_t \delta_t^i \nabla_{\phi^i} Q_{\text{SR}}^i(o_t^i, \boldsymbol{a_t}). \qquad (5)$$

We can further intuitively comprehend Eq. 3: when $w_t^{ij}$ attains the maximum value of $1/(N-1)$, $j$ has taken the best action that maximized $Q_{\text{SR}}^i$ and $i$ predicts with a probability of 1 that $j$ will select the action that minimizes $Q_{\text{SR}}^i$. At this point, the value $j$ brings to $i$ far exceeds $i$'s psychological expectation, which corresponds to the real-world scenario where people feel particularly happy when they are helped by someone they might have thought was unkind to them. So the amount of $i$'s gifting to $j$ reaches the maximum. It is worth noting that Eq. 3 may yield a negative value, indicating that the agent is required to acquire rewards from other agents. Dealing with this scenario becomes intricate, particularly when the other agent is uncooperative. As this type of competition is not the primary focus of our research, we assign $w^{ij} = 0$ when $w^{ij} \leq 0$.

### 4.3 Analysis in Iterated Matrix Game

We use iterated matrix games to theoretically analyze LASE's learning process. The iterative matrix game is to play multiple rounds of a single game with the payoff matrix shown in Tab. 1, where both players get a payoff of $R$ by mutual cooperation (C) and $P$ by mutual defection (D). If one player defects and the other cooperates, the defector receives a reward of $T$, while the cooperator receives a reward of $S$. We normalize $R$ to 1 and $S$ to 0, and let $0 \leq T \leq 2, -1 \leq S \leq 1$ which is shown to be sufficient to characterize the three typical kinds of dilemmas in Tab. 3 [26].

Table 1: Matrix-form game

| P1/P2 | C | D |
|---|---|---|
| C | $(R, R)$ | $(S, T)$ |
| D | $(T, S)$ | $(P, P)$ |

We carry out a closed-form gradient descent analysis on LASE in the two-player iterated matrix games and derive the policy update rule Eq. 15 and Eq. 16, where each agent $i$ optimizes the reward after gifting $r^{i,\text{tot}}$. The detailed deduction is provided in Appendix A. Then we simulate the policy update iteratively with random initial value and plot LASE's cooperation probability after convergence under various game parameters as illustrated in Fig. 2.

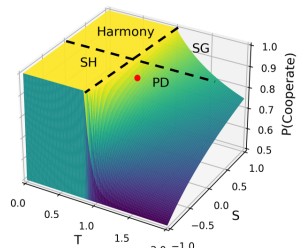

The results demonstrate that LASE converges to pure cooperation in Harmony and SH. In the more intense games, SG and PD, LASE stabilizes at cooperating with a probability greater than 0.5, successfully escaping from non-efficient Nash equilibria.

## 5 Experimental setup

### 5.1 Environments

**Iterated Prisoner's Dilemma (IPD).** Here, we use iterated prisoner's dilemma (IPD) as an illustration to validate the theoretical analysis of LASE conducted in Section 4.3 and Appendix A. The specific game parameters are set as $[R, S, T, P] = [1, -0.2, 1.2, 0]$, represented

Figure 2: The cooperation probability of LASE agents after convergence under different matrix-game parameters. The X-axis and Y-axis represent two parameters $T$ and $S$ respectively, where $T \in [0 : 0.02 : 2]$ and $S \in [-1 : 0.02 : 1]$.

as the red dot in Fig. 2. We employ the memory-1 IPD introduced in [6], with the state $s = [CC, CD, DC, DD, s_0]$ comprising the joint action in the previous round and the initial state $s_0$. The action space consists of two discrete actions - cooperation (C) and defection (D). Each episode lasts for 100 timesteps.

To evaluate the ability of LASE to address more complex environments, we study its performance in partially observable SSDs. SSDs extend matrix-form games in terms of space, time, and number of agents. Here, we study four specific SSDs: Coingame, Cleanup, Sequential Stag-Hunt (SSH), and Sequential Snowdrift Game (SSG) (Fig. 3). Schelling diagrams (see Fig. 10) of the four environments validate that they are appropriate extensions of representative game paradigms (a detailed analysis is given in Appendix B). Below is a detailed description of the four environments.

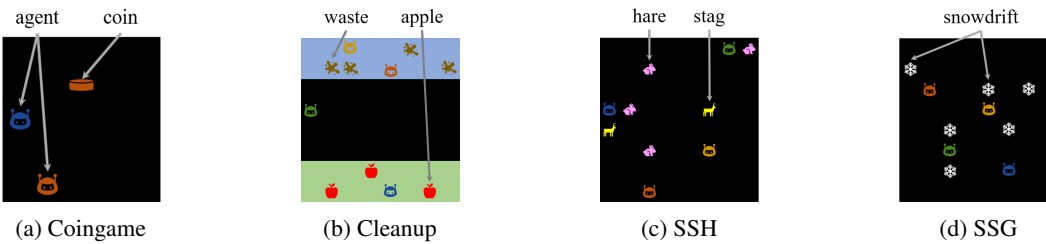

| (a) Coingame | (b) Cleanup | (c) SSH | (d) SSG |

Figure 3: Graphic representations of four SSDs: (a) Coingame (5×5 map), (b) Cleanup (8×8 map), (c) SSH (8×8 map), (d) SSG (8×8 map).

**Coingame.** The Coingame was originally introduced in [16] as a higher dimensional alternative to the IPD with multi-step actions. In this environment, two agents, 'red' and 'blue', collect coins in a $5 \times 5$ map. A coin is either red or blue and there is only one coin on the map at any timestep. Agents will get a reward of 1 by picking up a coin of any color. However, when an agent collects a coin of a different color, the other agent will lose 2 points. After a pickup, a new coin with a random color and random location appears immediately. Therefore, if each agent greedily picks up all the coins, the sum of their expected scores will be 0.

**Cleanup.** In Cleanup, the goal of the agent is to gather as many apples as possible, with each apple carrying a reward of +1. However, the accumulation of waste in the river steadily approaches a depletion threshold, causing a linear decline in the apple growth rate to 0. At the beginning of each episode, the waste level exceeds the threshold and there are no apples in the map. This places the agent in a social dilemma: while individually focusing on collecting apples under the map leads to higher rewards, if all agents opt to refuse waste cleanup, no rewards are obtained. To maintain consistency with other environments, we partially modify the setting of cleanup from [11]. We eliminate the actions of firing beams (cleaning and zapping) and require the agent to move to the waste's position to clean it. This does not alter the nature of the dilemma but makes it more challenging because the cleaning beam could have helped the agent clean from a distance.

**Sequential Stag-Hunt (SSH).** This environment is inspired by Markov Stag Hunt in [23]. Each agent can get a reward of +1 by hunting a hare while hunting stags is more challenging and requires two or more agents. Each stag can bring a reward of +10, which is divided equally among the agents that jointly hunted it. The agent is immediately removed from the environment after successfully hunting once. Therefore, if an agent chooses to hunt stags, it must contend with the risk of no one cooperating with it. In contrast, hunting rabbits is a safer choice.

**Sequential Snowdrift Game (SSG).** In SSG, there are 6 piles of snowdrifts which can be removed by the agent. A pile of removed snowdrifts brings a +6 reward for each agent, but the remover incurs a cost of 4. So the agent waiting for others to remove the snowdrift (free-rider) can obtain a higher return. However, if no one chooses to remove it, no rewards can be obtained by anyone.

## 5.2 Implementations

We employ fully decentralized training and execution for the agents, where all network parameters are independent. The policy structure of the agent comprises 2 convolutional layers for encoding observations, an LSTM layer to capture temporal information, and several fully connected layers activated by ReLU. The input is a multi-channel binary tensor, with the specific number of channels determined by the characteristics of different environments. For example, in Cleanup, 7 channels are

incorporated, wherein the first four channels denote the positions of the four agents, the fifth and sixth channels signify waste and apples, and the last channel distinguishes between the inside and outside of the map through masking. To ensure adequate exploration, we let $\widetilde{\pi}(a|o) = (1 - \epsilon)\pi(a|o) + \epsilon/|\mathcal{A}|$, with $\epsilon$ decaying linearly from $\epsilon_{\text{start}}$ to $\epsilon_{\text{end}}$ over $\epsilon_{\text{div}}$ episodes.

The SR policy network and SR value network of the SRI module share two convolutional layers, followed by their own linear layers activated by ReLU function. The difference is that the inputs of the linear layers in SR value network include an additional concatenated one-hot vector of the joint action. The observation conversion network uses a single convolutional layer to encode observation, which is then concatenated with a one-hot vector representing the ID of other agents. This concatenated input is then processed through two linear layers, with the resulting output normalized using the sigmoid function.

In IPD, we modify the implementation by removing convolutional layers, reducing the parameters of FC layers, and appropriately increasing the learning rate to adapt the algorithm to this environment. See Appendix C for more details about the implementations of LASE.

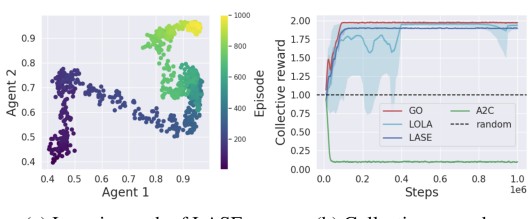

(a) Learning path of LASE  (b) Collective reward

Figure 4: Results in IPD. (a) The learning path of two LASE agents. They start from the lower left and converge to the upper right of the phase diagram, where both agents cooperate with a probability around 0.93. (b) The collective reward of LASE and baselines. Five seeds are randomly selected for the experiment. The solid line represents the mean performance, while the shaded area indicates the standard deviation.

## 5.3 Baselines

Independent advantage actor-critic labeled **A2C** [22] is a classical gradient-based RL algorithm. **LIO** [36] learns an incentive function through the learning updates of reward recipients. **LOLA** [6] considers the learning process of other agents when updating its own policy parameters. **IA** [11, 32] modifies the individual reward function by introducing inequity aversion. **SI** [13, 32] achieves coordination by rewarding agents for having causal influence over other agents' actions. We also show the approximate upper bound on performance by training the group optimal (**GO**) agents to maximize the collective reward. And we conduct ablation experiments **LASE w/o**, by removing the observation conversion network and replacing $\pi_{\text{SR}}^{i}(a_t^{j'}|\hat{o}_t^j)$ in Eq. 3 with $1/|\mathcal{A}^j|$.

## 6 Results

### 6.1 LASE promotes cooperation in social dilemmas

In **IPD**, as shown in Fig. 4, LASE successfully escapes the dilemma of non-efficient Nash Equilibrium (D, D). Both LASE agents converge to cooperate with a high probability, around 0.93 (see Fig. 4a), accompanied by a high collective reward. This is consistent with the theoretical results shown in Fig. 2, validating the effectiveness of LASE in dealing with social dilemmas. LASE is better at convergence speed and stability than LOLA, which also achieves a high collective reward. Unsurprisingly, GO, aiming to maximize group reward, reaches the upper bound. On the other hand, A2C, optimizing for one's own return, easily falls into the Nash equilibrium, where everyone defects and the group reward reduces to the minimum.

In **SSH** and **SSG**, LASE nearly reaches the upper bound of the total rewards as shown in Fig. 5a and Fig. 5b: a total reward of 20 for successfully hunting two stags in SSH and a total reward of 120 in SSG for removing all the six snowdrifts. In **Coingame** (Fig. 5c) and **Cleanup** (Fig. 5d), LASE exhibits commendable performance by effectively avoiding the sub-optimal equilibrium, where everyone defects. GO outperforms LASE in SSG, Coingame, and Cleanup, because its optimization objective of maximizing the collective reward of all the agents and a strong assumption of the accessibility of everyone's reward function can help it avoid the dilemma.

On the other hand, it's worth emphasizing that the architecture of GO also gets it into the lazy problem [31], shown as its underperformance in SSH (Fig. 5a). In SSH, early hunting leads to moving

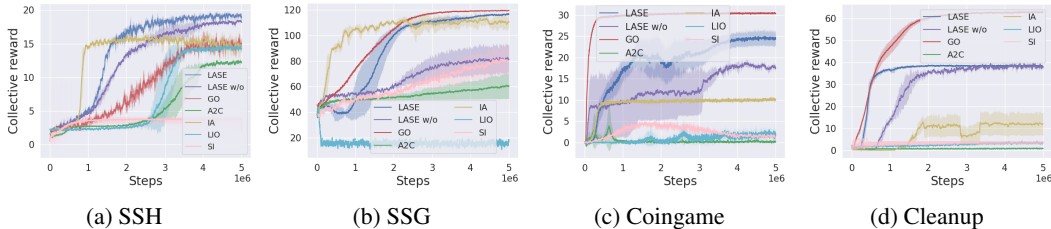

| (a) SSH | (b) SSG | (c) Coingame | (d) Cleanup |

Figure 5: Learning curves in four SSDs. Shown is the collective reward. All the curves are plotted using 5 training runs with different random seeds, where the solid line is the mean and the shadowed area indicates the standard deviation.

out of the environment and failing to obtain the group rewards of others' later hunting. Thus, GO may not hunt until the last few steps, and likely misses the the opportunity of cooperating to hunt stags.

A2C and SI hardly improve the collective return in Coingame and Cleanup, and fail to outperform LASE in SSH and SSG, where the dilemma is less intense. Although IA can directly access the rewards of others and adjust its own intrinsic rewards accordingly, it is still unable to get a higher collective return than LASE. Also as a gifting algorithm, LIO struggles to perform well in the four environments. Meanwhile, the need to manually specify the hyperparameters used to scale the amount of gifting also creates challenges for LIO to apply to different environments. In contrast, LASE breaks the limitations and achieves significantly better performance than LIO.

LASE w/o's convergence speed and performance are affected to a certain extent but not significantly. This is because LASE w/o differs from LASE by replacing the policy predictions of others with uniform policy to compute counterfactual baselines. It means that the gifting will continue as long as the first term of the numerator is substantial in Eq. 3. On the other hand, LASE builds a counterfactual baseline dynamically through perspective taking, and only gifts when the co-player's real action is superior to the baseline. Tab. 2 shows the mean of the gifting weights to other agents over the last 1e4 episodes of training, showing that LASE's gifting weights are lower than those of LASE w/o. Regarding gifting as a form of communication, LASE is valuable in reducing communication costs. Furthermore, to test LASE's scalability, we implement LASE in the extended Cleanup and SSG with 8 agents and a larger map. Results show that LASE is able to deal with more complex environments (see details in Section D.1).

Table 2: Mean of the gifting weights for LASE and LASE w/o during the last 10000 episodes of self-play training.

|  | Coingame | Cleanup | SSG | SSH |
|---|---|---|---|---|
| LASE | **0.222** | **0.346** | **0.234** | **0.072** |
| LASE w/o | 0.343 | 0.457 | 0.354 | 0.081 |

## 6.2 LASE promotes fairness

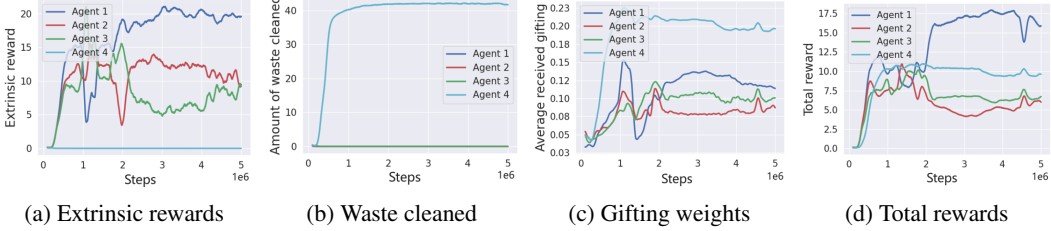

| (a) Extrinsic rewards | (b) Waste cleaned | (c) Gifting weights | (d) Total rewards |

Figure 6: Learning curves of each agent in Cleanup. Since the division of labor for different random seeds is not the same, only the results under one seed are shown to distinguish individual performance.

Considering the fact that the division of labor in Cleanup is more pronounced than in other environments and that cooperation in Cleanup is purely altruistic, making the dilemma more challenging, we take Cleanup as an example to show LASE's ability to promote fairness. Fig. 6a and Fig. 6b show the extrinsic rewards of each agent and the amount of waste cleaned by each in Cleanup. We find Agent 4 is the only one to clean and does not receive any extrinsic reward.

Fig. 6c illustrates the gifting weights each agent received from the other three agents. Agent 4 gets the most gifts, indicating that its cleaning contribution to the team is recognized and rewarded by the other three agents. Fig. 6d shows reward curves after gifting, where the reward gap between Agent 4 and the other three agents shrinks, implying fairness within the group is improved.

Agents 1-3 always collect apples without cleaning, but this is not a free-rider behavior. Both cleaning and collecting are indispensable to get rewards in Cleanup. Although Agent 4 does not obtain any reward directly from the environment, the acquired reward is redistributed through gifting, narrowing the gap of reward between the cleaner and collectors. In contrast, although GO obtains higher collective returns, it sacrifices the individual interests of Agent 1 and Agent 2, while only Agent 3 and Agent 4 can get rewards Fig. 7.

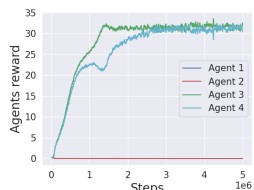

Figure 7: Four GOs' rewards in Cleanup.

We use *Equality (E)* given by $E = 1 - \frac{\sum_{i=1}^{N} \sum_{j=1}^{N} |R_i - R_j|}{2N \sum_{i=1}^{N} R_i}$ [20] to quantify the fairness, where the second term is the Gini inequality index. The greater the value of $E$, the fairer it is. We can get $E(\text{LASE}) \approx 0.802$, $E(\text{GO}) \approx 0.496$, showing that LASE achieves a higher level of fairness.

### 6.3 LASE distinguishes co-players and responds adaptively

To evaluate LASE's adaptation ability to interact with various types of agents, we conduct an experiment in which a focal LASE agent interacts with three rule-based agents: cooperator (always clean up waste), defector (always try to collect apples), and a random agent. The gifting weights of LASE to the other agents are shown in Fig. 8. LASE can explicitly distinguish between different types of co-players. Moreover, it responds in a manner that aligns with human values: preferring to share rewards with cooperators rather than defectors.

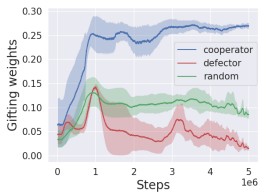

Figure 8: LASE's gifting weights to three rule-based co-players.

To study how LASE responds dynamically and how it affects collective behavior, we conduct an experiment where one focal LASE agent interacts with three A2C agents (Background agents, Bgs). A GO agent is trained in the same way for comparison with LASE. Fig. 9a displays each agent's rewards after training for 30k episodes, whereas the LASE group shows the reward after gifting.

LASE and GO improve the group reward to a similar level, while four A2C agents will converge to the equilibrium of defection and gain almost no reward (see Fig. 5d). With gifting, LASE incentives *Bg3* to clean as shown in Fig. 9b. Thus, LASE and the other two *Bg*s can get rewards by gathering apples. But for GO, the focal agent sacrifices itself to undertake all the cleaning tasks and cannot get any reward.

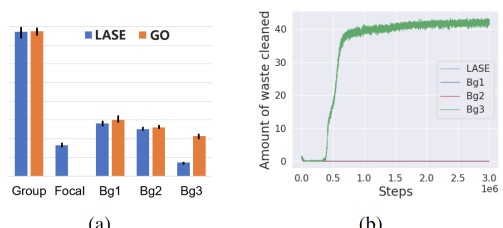

Figure 9: An LASE (or GO) agent interacts with three A2C agents in Cleanup. (a) The average reward for LASE and GO groups after training 30k episodes. The LASE group shows the reward after gifting. The first two bars show whole groups' rewards. The remaining bars show the average reward for each agent. (b) The amount of the waste cleaned by each agent in the LASE group.

GO and LASE represent two different methods to foster cooperation. GO sacrifices its own interest to promote collective reward. LASE attempts to alleviate social dilemmas by incentivizing others to cooperate. When such an incentive mechanism fails, LASE will no longer gift those agents who constantly exploit it, like the defector in Fig. 8. Overall, we believe that LASE is a more efficient and secure policy for SSDs, as it can promote cooperation as well as avoid potential exploitation by others.

## 7 Conclusion

We introduce LASE, a decentralized MARL algorithm that fosters cooperation through gifting while safeguarding individual interests in mixed-motive games. LASE uses counterfactual reasoning to

infer the social relationships with others which captures the influences of others' actions on LASE and modulates the gifting strategy empathetically. In particular, to empower LASE with the ability to infer others' policies in partially observable and decentralized environments, we establish a perspective taking module for LASE. Both theoretical analyses in matrix-form games and experimental results across diverse SSDs show that LASE can effectively promote cooperative behavior while ensuring relative fairness within the group. Furthermore, LASE is also able to recognize various types of co-players and adjust its gifting strategy adaptively to avoid being exploited, enabling broad applicability in complex real-world multi-agent interactions, such as automated negotiations in E-commerce and decision-making in autonomous driving. Whilst LASE exhibits superior abilities, there are some limitations of our method. What would be the consequence of giving agents the ability to refuse gifts? How to extend the algorithm LASE to continuous action space? These problems will illuminate our future work.

## Acknowledgments and Disclosure of Funding

This work is supported by the National Science and Technology Major Project (No. 2022ZD0114904).

This work is also supported by the project "Wuhan East Lake High-Tech Development Zone, National Comprehensive Experimental Base for Governance of Intelligent Society".

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

## A Analysis in Iterated Matrix Games

For a general iterated matrix game whose payoff matrix is computed according to Tab. 1 at each step, we can let $\theta^i$ for $i \in \{1, 2\}$ denote each agent's probability of taking the cooperative action, and let $\hat{\theta}^j$ for $j \in \{2, 1\}$ denote each agent's prediction of the other's policy. To avoid the difficulties caused by the coupled update of reinforcement learning for the theoretical analysis, we make the simplifying assumption that the three networks in SRI module have been fully trained. We further assume that this two-player game is fully observable, and we can make the following approximation to Eq. 3:

$$
w_t^{ij} = \frac{1}{N-1} \frac{Q^i(o_t^i, \boldsymbol{a_t}) - \sum_{a_t^{j'}} \pi_{\text{sc}}^i(a_t^{j'} | \hat{o}_t^j) Q^i(o_t^i, (\boldsymbol{a_t^{-j}}, a_t^{j'}))}{\max_{a_t^{j'}} Q^i(o_t^i, (\boldsymbol{a_t^{-j}}, a_t^{j'})) - \min_{a_t^{j'}} Q^i(o_t^i, (\boldsymbol{a_t^{-j}}, a_t^{j'}))}
\tag{6}
$$

$$
\approx \frac{1}{N-1} \frac{r^i(a^i, a^j) - (\hat{\theta}^j r^i(a^i, C) + (1 - \hat{\theta}^j) r^i(a^i, D))}{r^i(a^i, C) - r^i(a^i, D)}
$$

Here, $r^i(a^i, a^j)$ represents player $i$'s reward determined by the payoff matrix which only relies on the two players' actions without state or observation. The update rule Eq. 5 states that it is feasible to replace $Q^i$ by $r^i$. Then we can calculate the gifting weights under the four combinations of actions: CC, CD, DC, and DD. We take $w^{12}$ for example:

$$
w_{CC}^{12} = \frac{r^1(C, C) - (\hat{\theta}^2 r^1(C, C) + (1 - \hat{\theta}^2) r^1(C, D))}{r^1(C, C) - r^1(C, D)} = \frac{R - (\hat{\theta}^2 \cdot R + (1 - \hat{\theta}^2 \cdot S))}{R - S} = 1 - \hat{\theta}^2
\tag{7}
$$

$$
w_{CD}^{12} = \frac{r^1(C, D) - (\hat{\theta}^2 r^1(C, C) + (1 - \hat{\theta}^2) r^1(C, D))}{r^1(C, C) - r^1(C, D)} = \frac{S - (\hat{\theta}^2 \cdot R + (1 - \hat{\theta}^2 \cdot S))}{R - S} = -\hat{\theta}^2
\tag{8}
$$

$$
w_{DC}^{12} = \frac{r^1(D, C) - (\hat{\theta}^2 r^1(D, C) + (1 - \hat{\theta}^2) r^1(D, D))}{r^1(D, C) - r^1(D, D)} = \frac{T - (\hat{\theta}^2 \cdot T + (1 - \hat{\theta}^2 \cdot P))}{T - P} = 1 - \hat{\theta}^2
\tag{9}
$$

$$
w_{DD}^{12} = \frac{r^1(D, D) - (\hat{\theta}^2 r^1(D, C) + (1 - \hat{\theta}^2) r^1(D, D))}{r^1(D, C) - r^1(D, D)} = \frac{P - (\hat{\theta}^2 \cdot T + (1 - \hat{\theta}^2 \cdot P))}{T - P} = -\hat{\theta}^2
\tag{10}
$$

Similarly as Section 4, we set $w^{12}$ less than 0 to 0. So agent 1's reward distribution scheme is:

$$
\begin{aligned}
r_{CC}^{12} &= w_{CC}^{12} R = (1 - \hat{\theta}^2) R, r_{CC}^{11} = R - r_{CC}^{12} = \hat{\theta}^2 R, \\
r_{DC}^{12} &= w_{DC}^{12} T = (1 - \hat{\theta}^2) T, r_{DC}^{11} = T - r_{DC}^{12} = \hat{\theta}^2 T, \\
r_{CD}^{12} &= r_{DD}^{12} = 0, r_{CD}^{11} = S, r_{DD}^{11} = P
\end{aligned}
\tag{11}
$$

Agent 2's computation is symmetric. The total reward received by each agent is

$$
r^{1,\text{tot}} = [\hat{\theta}^2 R + (1 - \hat{\theta}^1) R, S + (1 - \hat{\theta}^1) T, \hat{\theta}^2, P]
\tag{12}
$$

$$
r^{2,\text{tot}} = [\hat{\theta}^1 R + (1 - \hat{\theta}^2) R, \hat{\theta}^1, S + (1 - \hat{\theta}^2) T, P]
\tag{13}
$$

The value function for each agent is defined by

$$
V^i(\theta^1, \theta^2) = \sum_{t=0}^{\infty} \gamma^t p^T r^{i,\text{tot}},
\tag{14}
$$

$$
\text{where } p = [\theta^1 \theta^2, \theta^1(1 - \theta^2), (1 - \theta^1)\theta^2, (1 - \theta^1)(1 - \theta^2)].
$$

Agent 2 updates its policy by

$$
\begin{aligned}
\theta^2 &= \theta^2 + \alpha \nabla_{\theta^2} V^2(\theta^1, \theta^2) \\
&= \theta^2 + \frac{\alpha}{1 - \gamma} \nabla_{\theta^2} \left\{ \theta^1 \theta^2 \left[ \hat{\theta}^1 R + (1 - \hat{\theta}^2) r \right] \right. \\
&\quad \left. + \theta^1 (1 - \theta^2) \hat{\theta}^1 T + \theta^2 (1 - \theta^1) \left[ S + (1 - \hat{\theta}^2) \right] + (1 - \theta^1)(1 - \theta^2) P \right\} \\
&= \theta^2 + \frac{\alpha}{1 - \gamma} \left\{ \left[ \hat{\theta}^1 R + (1 - \hat{\theta}^2) R - \hat{\theta}^1 T \right] \theta^1 + \left[ S + (1 - \hat{\theta}^2) T - P \right] (1 - \theta^1) \right\}
\end{aligned}
\tag{15}
$$

By symmetry, agent 1 updates its policy by

$$\theta^1 = \theta^1 + \frac{\alpha}{1-\gamma}\left\{\left[\hat{\theta}^2 R + (1-\hat{\theta}^1)R - \hat{\theta}^2 T\right]\theta^2 + \left[S + (1-\hat{\theta}^1)T - P\right](1-\theta^2)\right\} \quad (16)$$

We assume that LASE's prediction to the others is accurate when simulating in Section 4.3, i.e $\hat{\theta}^1 = \theta^1, \hat{\theta}^2 = \theta^2$. And we let $\alpha = 10^{-3}, \gamma = 0.99$.

## B Environments

### B.1 Validating the environments

In this section, we will show that our four environments are all *sequential social dilemmas* defined in [11]: An $N$-player sequential social dilemma is a tuple $(\mathcal{M}, \Pi = \Pi_c \sqcup \Pi_d)$ of a Markov game and two disjoint sets of policies, said to implement cooperation and defection respectively, satisfying the following properties. Consider the strategy profile $(\pi_c^1, \ldots, \pi_c^l, \pi_d^1, \ldots, \pi_d^m) \in \Pi_c^l \times \Pi_d^m$ with $l + m = N$. We shall denote the average payoff for the cooperating policies by $R_c(l)$ and for the defecting policies by $R_d(l)$. $(\mathcal{M}, \Pi)$ is a sequential social dilemma iff the following hold:

1. Mutual cooperation is preferred over mutual defection: $R_c(N) > R_d(0)$.
2. Mutual cooperation is preferred to being exploited by defectors: $R_c(N) > R_c(0)$.
3. Either the *fear* property, the *greed* property, or both:
   - Fear: mutual defection is preferred to being exploited. $R_d(i) > R_c(i)$ for sufficiently small $i$.
   - Greed: exploiting a cooperator is preferred to mutual cooperation. $R_d(i) > R_c(i)$ for sufficiently large $i$.

A Schelling diagram is a game representation that highlights interdependencies between agents, showing how the choices of others shape one's own incentives. It plots the curves $R_c(l+1)$ and $R_d(l)$ as shown in Fig. 10. All the environments satisfy the first two properties of sequential social dilemmas: $R_c(N) > R_d(0)$ and $R_c(N) > R_d(0)$. In SSH, fear promotes defection: $R_d(0) > R_c(0)$. In Cleanup and SSG, the problem is greed: $R_d(3) > R_c(3)$. Coingame suffers from both temptations to defect. This indicates that our experimental environments include all three different types of sequential social dilemmas corresponding to Tab. 3.

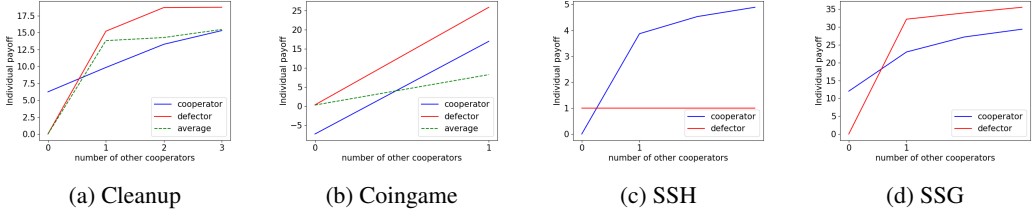

|  (a) Cleanup  |  (b) Coingame  |  (c) SSH  |  (d) SSG  |

Figure 10: The Schelling diagram of Cleanup, Coingame, Sequential Stag-Hunt (SSH) and Sequential Snowdrift Game (SSG). The dotted line shows the overall average return where the individual chooses defection.

Table 3: Classification of social dilemmas. In all dilemmas, mutual cooperation yields higher payoffs than mutual defection, yet each dilemma provides an incentive for defection. In the Snowdrift game, one player can gain a higher payoff by defecting when the other cooperates, while in the Stag Hunt game, a player can achieve higher payoffs by defecting when the other defects. The Prisoner's Dilemma encompasses both types of incentives.

| Social dilemmas | Abbreviation | Parameters |
|---|---|---|
| Snowdrift | SG | $T > 1 > S > 0$ |
| Stag Hunt | SH | $1 > T > 0 > S$ |
| Prisoner's Dilemma | PD | $T > 1 > 0 > S, 2 > T + S$ |

## B.2    Environment details

**IPD.** Each agent makes decisions based on the actions of the two players in the previous step, so this is a fully observable environment unlike other environments. We trained for 10k episodes, each with 100 steps.

**Coingame.** Map size is $5 \times 5$. Agent's action space is $\mathcal{A} = \{$up, down, left, right$\}$. The state is represented as a $5 \times 5 \times 5$ binary tensor. The five channels are $\{$blue agent, red agent, blue coin, red coin, mask$\}$, where the first two channels encode the location of each agent, the last channel distinguishes between parts within and beyond the boundary, and the other two channels encode the location of the coin if any exist. If two agents walk on the coin at the same time, one of them is randomly selected to successfully pick it up. Cooperative agents will only try to collect coins with their own color, while self-interested agents tend to greetingly collect all coins. Each episode lasts for 100 steps.

**Cleanup.** Map size is $8 \times 8$. $\mathcal{A} = \{$up, down, left, right, stay, clean, pick$\}$, where the last two actions require the agent to be in the same position as the waste or the apple. The seven channels are $\{$agent 1, ..., agent 4, waste, apple, mask$\}$, The parameters with the same meaning as the open-source implementation about Cleanup [32] are shown in Tab. 4. To achieve cooperation, agents need to take the initiative to undertake part of the cleaning task to help improve the group's revenue. A defecting agent will just keep waiting for the apple to grow and gather it. Each episode lasts for 100 steps.

Table 4: Environment parameters in Cleanup

| Parameter | Value |
| --- | --- |
| map_size | $8 \times 8$ |
| appleRespawnProbablity | 0.4 |
| wasteSpawnProbability | 0.5 |
| thresholdDepletion | 0.5 |
| thresholdRestoration | 0.0 |
| view_size | $5 \times 5$ |
| max_steps | 100 |

**SSH.** Map size is $8 \times 8$. $\mathcal{A} = \{$up, down, left, right, stay, hunt hare, hunt stag$\}$. Agents must be in the same position as the prey to hunt and the prey doesn't respawn after being hunted. The seven channels are $\{$agent 1, ..., agent 4, hare, stag, mask$\}$. Cooperative agents are happy to hunt deer with others, while defectors only hunt rabbits to avoid the risk of not getting a payoff. Each episode lasts for 30 steps.

**SSG.** Map size is $8 \times 8$. $\mathcal{A} = \{$up, down, left, right, stay, remove snowdrift$\}$. Agents must be in the same position as the snowdrift to remove it and the removed snowdrift doesn't respawn. The six channels are $\{$agent 1, ..., agent 4, snowdrift, mask$\}$. Cooperators will proactively remove snowdrifts to bring high rewards to the team, while defectors will just wait for others to remove them. Each episode lasts for 50 steps.

## C    Implementation

The pseudocode for the LASE algorithm is shown in Algorithm 1.

**SSDs.** The actor-critic model interacting with environments utilizes two cascaded CNNs to process input data, with a kernel size of 3, stride of size 1 and 16 / 32 output channels. This is connected to one fully connected layer of size 128 activated by ReLU, and an LSTM with 128 cells. The actor head and critic head are two separate fully-connected layers that output the softmax normalized action policy and a scalar value respectively. The implementation of the intrinsic policy network and value network in OM is basically the same, but due to the need to further judge the actions of others, joint action space is added to the value network input dimension. In the observation transformation network, the input data is passed through a CNN with a kernel of size 3, stride of size 1 and 16 output channels, and is concatenated with a one-hot vector representing the agent index to input in two FC

---

**Algorithm 1** Learning to balance Altruism and Self-interest based on Empathy (LASE)

---

Initialize action policy $\pi^i$, SR policy network $\pi^i_{SR}$, SR value newtork $Q^i_{SR}$ and observation conversion network parameterized by $\theta^i, \mu^i, \phi^i, \eta^i$ and trajectory buffer $\mathcal{B}^i$, for each agent $i$
**for** $eps = 1$ to $max\_episodes$ **do**
    All agents interact with the environment for many steps, each agent $i$ gets a trajectory $\tau^i = (o^i_t, \boldsymbol{a_t}, r^{i,\text{env}}_t, o^i_{t+1})$ and stores it in $\mathcal{B}^i$
    **for** each agent $i$ **do**
        **for** each other agent $j$ **do**
            get $j$'s simulated observation $\hat{o}^j$ by $i$'s observation conversion network
            estimate $j$'s policy $\pi^i_{SR}(a^{j'}|\hat{o}^j)$ with SR policy network
            **for** $j$'s all possible actions $a^{j'}$ **do**
                compute $i$'s Q-value $Q^i_{SR}(o^i_t, (\boldsymbol{a_t^{-j}}, a^{j'}_t))$, with $j$ taking all possible actions $a^{j'}_t$ while the other agents' actions $\boldsymbol{a}^{-j}$ fixed
            **end for**
            compute the counterfactual baseline $\sum_{a^{j'}_t} \pi^i_{SR}(a^{j'}_t|\hat{o}^j_t) Q^i_{SR}(o^i_t, (\boldsymbol{a_t^{-j}}, a^{j'}_t))$
            compute gifting weights $w^{ij}$ by Eq. 3
        **end for**
        get $w^{ii}$ by $w^{ii} = 1 - \sum_{j=1, j\neq i}^N w^{ij}$
    **end for**
    get $\boldsymbol{r}^{\text{tot}} \leftarrow \boldsymbol{r}$ and update policy $\pi^i$ to maximize the accumulated $\boldsymbol{r}^{\text{tot}}$
    **if** $eps$ mod $update\_frequency = 0$ **then**
        **for** each agent $i$ **do**
            sample a minibatch from $\mathcal{B}$, update $\mu^i, \phi^i, \eta^i$ by Eq. 4, Eq. 5
        **end for**
    **end if**
**end for**

---

layers of size 128. The output is reshaped to be the same size as the input observations. We use Adam optimizer [14] for all modules' training.

**IPD.** All the CNNs in IPD are removed. The size of FC layer and the cell number in LSTM are scaled down to 32.

Table 5: Hyperparameters

(a) Hyperparameters in SSDs

| Parameter | Value | Parameter | Value |
|---|---|---|---|
| $\epsilon_{\text{start}}$ | 0.5 | $\alpha_\theta$ | 1e-4 |
| $\epsilon_{\text{div}}$ | 2e3 | $\alpha_\mu$ | 3e-5 |
| $\epsilon_{\text{end}}$ | 0.05 | $\alpha_\phi$ | 3e-5 |
| $\gamma_{\text{sc}}$ | 0.98 | $\alpha_\eta$ | 5e-5 |
| $\gamma$ | 0.98 | update_freq | 20 |
| $\delta$ | 0.1 | batch_size | 1000 |

(b) Hyperparameters in IPD

| Parameter | Value | Parameter | Value |
|---|---|---|---|
| $\epsilon_{\text{start}}$ | 0.5 | $\alpha_\theta$ | 5e-3 |
| $\epsilon_{\text{div}}$ | 1e3 | $\alpha_\mu$ | 1e-3 |
| $\epsilon_{\text{end}}$ | 0.01 | $\alpha_\phi$ | 1e-3 |
| $\gamma_{\text{sc}}$ | 0.98 | $\alpha_\eta$ | 1e-3 |
| $\gamma$ | 0.95 | update_freq | 20 |
| $\delta$ | 0.1 | batch_size | 64 |

**Experiments Compute Resources**

CPU: 128 Intel(R) Xeon(R) Platinum 8369B CPU @ 2.90GHz; Total memory: 263729336 kB GPU: 8 NVIDIA GeForce RTX 3090; Memory per GPU: 24576 MiB The main experiments are as shown in Fig. 5. An experiment takes about 2000 MiB of the GPU and takes about 1.5 days to run. About 5-7 experiments can be run simultaneously on the machine. Due to the need to debug and adjust parameters, approximately double the amount of computation is required.

## D  Additional Results

### D.1  Scalability of LASE

To test LASE's scalability, we have extended Cleanup and Snowdrift as Tab. 6.

Table 6: Environmental parameters of extended Cleanup and SSG

|              | Map Size        | Player Num    | Obs Size      | Init Waste/ Snowdrift num | Episode len        |
| ------------ | --------------- | ------------- | ------------- | ------------------------- | ------------------ |
| Cleanup.Extn | $8 \to 12$      | $4 \to 8$     | $5 \to 7$     | $8 \to 16$                | $100 \to 150$      |
| SSG.Extn     | $8 \to 12$      | $4 \to 8$     | $5 \to 7$     | $6 \to 12$                | $50 \to 70$        |

Tab. 7 shows the self-play results of LASE and baselines. LASE outperforms baselines in the two more complex environments.

Table 7: Self-play results (total reward) in extended Cleanup and SSG

| Total reward | LASE    | IA      | LIO     | SI      | A2C     |
| ------------ | ------- | ------- | ------- | ------- | ------- |
| Cleanup.Extn | 56.513  | 20.798  | 1.294   | 3.548   | 0.135   |
| SSG.Extn     | 232.564 | 227.762 | 20.317  | 207.461 | 134.964 |

### D.2  Estimate the uncertainty of their social relationship

We select the $w^{ij}$ data from the last $10^6$ timesteps of training to calculate their mean value and standard deviation, which estimates the uncertainty of social relationships. The calculation method is as follows:

$$\overline{w}^{ij} = \frac{\sum_{t=T_{\max}}^{T_{\max}-10^6} w_t^{ij}}{10^6}, \overline{w} = \sum_i^n \sum_{j,j\neq i}^n \overline{w}^{ij}, s = \frac{\sum_i^n \sum_{j,j\neq i}^n \sqrt{\frac{\sum_{t=T_{\max}}^{T_{\max}-1e6}(w_t^{ij}-\overline{w}^{ij})^2}{1e6-1}}}{n \times (n-1)}$$

We conduct a comparative experiment to replace the input of SR policy network $\hat{o}^j$ with $j$'s real observation $o^j$ . Here is the results:

The results show that the mean value of LASE's inferred social relationships closely match actual observations when available, although partial observability significantly increases uncertainty. Considering that the social relationships between people in real life tend to be relatively stable and do not change drastically, we think that a possible solution to handle the uncertainty of social relationships is to introduce some smoothing techniques for $w^{ij}$ to reduce the variance of social relationships over time. This approach will be explored in our future work.

Meanwhile, It is important to note that Figure 8 and corresponding analysis show that LASE is able to correctly infer the relationships with different co-players and respond properly. Specifically, in the experiments conducted in Section 6.3, one LASE interacts with three rule-based co-players: cooperator, defector and random. The results show the $w^{ij}$ given to cooperative co-player is the

Table 8: The uncertainty of inferred social relationships

| $\overline{w}, s$   | SSH                   | SSH                   | Coingame              | Cleanup               |
| ------------------- | --------------------- | --------------------- | --------------------- | --------------------- |
| LASE w/o $o^j$      | $0.07184 \pm 0.01921$ | $0.02341 \pm 0.00939$ | $0.22243 \pm 0.18594$ | $0.34572 \pm 0.01509$ |
| LASE w/ $o^j$       | $0.06278 \pm 0.01136$ | $0.03157 \pm 0.00527$ | $0.19317 \pm 0.05493$ | $0.29465 \pm 0.00889$ |

largest, significantly higher than that given to the other two co-players. The $w^{ij}$ given to random co-player comes second, and the smallest is given to defector. These results demonstrate the consistency between LASE's estimates of the relations and the ground truth.

### D.3 Compare the *Equality* with other baselines

As an evaluation metric, fairness should be evaluated alongside reward to measure algorithm performance effectively. Some algorithms may fail to address decision-making issues in mixed-motive games where each agent receives a small reward, but the reward disparity between agents is minimal, resulting in high fairness. Clearly, these methods are not effective. An effective method should both maximize group reward and ensure intra-group equity. Here, we include the fairness results of other baselines:

Table 9: All Equality results

| Fairness | SSH | SSG | Coingame | Cleanup |
|---|---|---|---|---|
| LASE | 0.994 | 0.951 | 0.835 | 0.802 |
| LASE w/o | 0.986 | 0.862 | 0.848 | 0.685 |
| GO | 0.968 | 0.856 | 0.785 | 0.496 |
| IA | 0.984 | 0.877 | 0.898 | 0.708 |
| LIO | 0.931 | 0.985 | 0.745 | 0.545 |
| SI | 0.995 | 0.937 | 0.750 | 0.892 |
| A2C | 0.997 | 0.854 | 0.831 | 0.824 |

## E  Broader Impact

The rapid advancement of AI technology has brought about an explosion in the number of agents, making it unfeasible to rely on a centralized controller to achieve coordinated collective behavior. The question of how to design independent agents that excel in specific tasks and can demonstrate adequate social behavior when interacting with humans or other agents remains an open challenge. We take a tentative step towards this problem by introducing the mechanism of gifting and the theory of empathy in human society. We believe our work will have a positive impact on the interaction of multi-agent systems, the alignment of AI with human values, and the development of AI safety.

