# OpenReview forum: "Learning to Balance Altruism and Self-interest Based on Empathy in Mixed-Motive Games"
_NeurIPS.cc/2024/Conference — NeurIPS 2024 poster_

### Official Review · Reviewer_vMEk · 2024-07-11

**Soundness:** 3
**Presentation:** 3
**Contribution:** 2
**Rating:** 5
**Confidence:** 4

**Summary:**

The paper introduces LASE, a novel distributed multi-agent reinforcement learning algorithm. LASE aims to foster altruistic cooperation through a gifting mechanism while avoiding exploitation in mixed-motive games. The algorithm dynamically adjusts the allocation of rewards based on social relationships inferred using counterfactual reasoning. The paper reports comprehensive experiments in various mixed-motive games, demonstrating that LASE effectively promotes group collaboration and adapts policies to different co-player types without compromising fairness.

**Strengths:**

The paper presents an innovative combination of gifting mechanisms and counterfactual reasoning within a multi-agent reinforcement learning framework. This approach to dynamically adjust reward allocation based on inferred social relationships is novel and well-grounded in developmental psychology theories of empathy.

The paper is clearly written and well-organized. The methodology and experimental setup are described in detail, making it relatively easy to follow and reproduce the results. Additionally, the authors conducted comprehensive experiments across various mixed-motive game scenarios, thoroughly demonstrating the effectiveness of LASE in promoting cooperation and fairness. The breadth and depth of these experiments add significant credibility to the proposed method.

**Weaknesses:**

The proposed method, while interesting, could be seen as incremental since it primarily combines existing techniques (gifting and counterfactual reasoning). The novelty could be better highlighted by contrasting more explicitly with prior works.

The more important weakness is that the paper lacks a detailed analysis of the individual contributions of the gifting mechanism and counterfactual reasoning. For example, there is no direct comparison with the original gifting method or an ablation study isolating the impact of the counterfactual reasoning module. This makes it difficult to discern the specific roles and contributions of these components to the overall performance.

Thus, I think the paper is below the acceptance bar for NeurIPS.

**Questions:**

1.How does LASE compare to the original gifting methods? Including a direct comparison or an ablation study focusing on the gifting mechanism could provide deeper insights into its contribution.
2.Can you provide an ablation study that isolates the impact of the counterfactual reasoning module? This would help in understanding its specific role and effectiveness.
3. Why not compare the Equality with other baseline methods? I am curious about the equality performance of baseline methods? from the report of their paper, I guess some of them will have not bad performance.

**Limitations:**

The paper seems lack of discussion about limitations and broader societal impacts.

---

> ### Author Rebuttal · Authors · 2024-08-07
>
> Thank you for your valuable comments on our work!
>
> > The comparison to the original gifting methods and the individual contributions of the gifting mechanism.
>
> Our primary contribution lies in designing an algorithm that can dynamically adjust the gifting strategy to different co-players, performing exceptionally well in multi-agent decision-making and adaptation. In contrast, most of the previous works predefine fixed gifting amounts, lacking adaptation ability. LIO and LToS [4] have learned gifting strategies, but underperform in mixed-motive scenarios.
>
> As far as we know, [1] first used the gifting mechanism to promote cooperation in sequential social dilemmas. The implementation of the gifting in [1] is to equip the agent with the extra action of a "gifting beam", which gives other agents in the beam a reward of $g$, and incurs an immediate penalty of $−g$ to itself. We try to approximatively implement this method by adding an additional gifting action to the A2C agents. However, since this action incurs an immediate $−g$ penalty, even a small $g$ will quickly reduce the probability of choosing this action, and the result is almost the same as that of A2C without gifting in our baselines which underperforms LASE.
>
> There are also some works that focus on how social dilemmas can be resolved by pre-defining the proportion of rewards to be distributed (gifted) [2] [3]. At one extreme, equal distribution makes everyone's goal the group's total benefit, the same as our GO baseline. For a clearer comparison, we conduct an additional ablation experiment using the converged gifting weight (as shown in Table 2) as a fixed gifting weight for each agent.
>
> ||SSH|SSG|Cleanup|Coingame|
> |---|---|---|---|---|
> |Original gifting|12.937|80.347|0.172|0.209|
> |A2C|12.476|81.285|0.274|0.036|
> |Fixed weight|14.626|55.117|35.704|3.435|
> |LASE|18.948|117.784|38.736|33.467|
>
> LASE outperforms because it can encourage cooperative behavior by rewarding some specific agents. In addition, LASE can dynamically adjust the proportion of rewards shared with different agents, which helps avoid being exploited by others.
>
> To sum up, our contribution to the gifting mechanism is mainly reflected in the fact that LASE can adjust the gifting strategy dynamically by inferring the social relationships with others, and can effectively promote group cooperation and avoid exploitation under various sequential social dilemmas.
>
> [1] Lupu et al., Gifting in multi-agent reinforcement learning, AAMAS 2020
>
> [2] Wang, et al., Emergent prosociality in multi-agent games through gifting, IJCAI 2021
>
> [3] Willis et al., Resolving social dilemmas through reward transfer commitments, ALA 2023
>
> > Ablation study, contribution and effectiveness about the counterfactual reasoning module.
>
> The social relationships inferred through counterfactual reasoning guide our gifting strategy and form the core of our algorithm, making ablation experiments on this module challenging. A feasible method is to use an end-to-end trained neural network to determine gifting weights, as done by LToS [4]. So we add LToS as one baseline:
>
> ||SSH|SSG|Cleanup|Coingame|
> |---|---|---|---|---|
> |LToS|12.476|77.386|1.912|-0.078|
> |LASE|18.948|117.784|38.736|33.467|
>
> It shows that the counterfactual reasoning module helps LASE outperform LToS. Meanwhile, LIO as a method that does not use counterfactual reasoning but directly uses neural networks to output gifted rewards has also been compared as a baseline in the paper.
>
> Counterfactual reasoning is an effective idea in multi-agent learning, however, most previous research has focused on cooperative tasks using the CTDE framework [5] or competitive tasks using counterfactual regret minimization (CFR) [6]. In contrast, LASE employs counterfactual reasoning to infer social relationships in a decentralized manner, achieving excellent performance in mixed-motive games. We believe this innovation has the potential to significantly enhance community development.
>
> [4] Yi, et al., Learning to share in multi-agent reinforcement learning, NeurIPS 2022
>
> [5] Foerster et al., Counterfactual Multi-Agent Policy Gradients, AAAI 2018
>
> [6] Zinkevich, et al., Regret minimization in games with incomplete information, NeurIPS 2007
>
> > Compare the Equality with other baseline methods.
>
> As an evaluation metric, fairness should be evaluated alongside reward to measure algorithm performance effectively. Some algorithms may fail to address decision-making issues in mixed-motive games where each agent receives a small reward, but the reward disparity between agents is minimal, resulting in high fairness. Clearly, these methods are not effective. An effective method should both maximize group reward and ensure intra-group equity. Thus, we compared only LASE and GO, which achieve the highest rewards. We have now included the fairness results of other baselines:
>
> |Fairness|SSH|SSG|Coingame|Cleanup|
> |---|---|---|---|---|
> |LASE|0.994|0.951|0.835|0.802|
> |LASE w/o|0.986|0.862|0.848|0.685|
> |GO|0.968|0.856|0.785|0.496|
> |IA|0.984|0.877|0.898|0.708|
> |LIO|0.931|0.985|0.745|0.545|
> |SI|0.995|0.937|0.750|0.892|
> |A2C|0.997|0.854|0.831|0.824|
>
> As you guess, LASE does not always outperform the baseline on fairness metrics, such as SI and A2C in SSH, LIO in SSG, IA in Coingame, and SI in Cleanup. However, LASE can significantly enhance group benefits while maintaining high fairness, which other baselines can’t.
>
> > Limitations and broader societal impacts.
>
> We have mentioned the limitations of our work in Section 7, Conclusion, including the assumption that each agent's reward function will be modified by other agents' gifts, whereas in reality, people may refuse gifts. Additionally, LASE currently focuses on finite discrete action spaces, and extending it to continuous action spaces is our next goal.
>
> We discussed the broader impact of our work in Appendix D. If you have any further questions, please feel free to ask! We are happy to discuss this with you!

---

> > ### Comment · Reviewer_vMEk · 2024-08-10
> > **Main concerns are addressed.**
> >
> > Dear Authors,
> >
> > Thanks for the additional experiments provided. The most of my concerns are addressed. And I suggest the authors to add the new experiments in the future version.  And I will increase my score.

---

> > > ### Author Response · Authors · 2024-08-11
> > > **Thanks for the response**
> > >
> > > Thank you very much for your valuable feedback and for taking the time to review our submission! We truly appreciate your insights and suggestions, which have helped us identify areas for improvement. We will carefully consider your comments and add the experiments in our revised version!

---

### Official Review · Reviewer_AnKj · 2024-07-11

**Soundness:** 2
**Presentation:** 3
**Contribution:** 2
**Rating:** 6
**Confidence:** 3

**Summary:**

This work proposes LASE, a multi-agent reinforcement learning framework that aims to improve co-operation between agents in mixed-motive games using transfer of rewards between agents in a zero-sum manner. Specifically, each agent uses counterfactual reasoning to compute a social relationship metric that computes the effect of their actions on the Q-values of other agents. Notably, LASE uses fully decentralized training, in contrast to many related works in the area. Finally, LASE outperforms existing baselines in a variety of popular mixed-motive environments such as Cleanup.

**Strengths:**

1.	The paper is well-organized, clearly written and technically sound. The general flow of the paper is smooth and proposed methods are explained reasonably well. The paper has an appropriate number of citations and properly details existing work in the related work section.
2.	The presented framework is fully decentralized, which gives it an advantage to most works in the area that use centralized training decentralized execution (CTDE).
3.	The results are generally promising with LASE showing significant gains in most environments tested. The analysis of co-operation strategies learned by different methods is interesting. In particular, I liked the GO vs LASE BG ablation study in Section 6.3.

**Weaknesses:**

1) The framework is relatively complex, as it requires learning two additional networks (perspective-taking and Q networks) for each agent.
2) Given the dependence of the method on joint action, I have doubts about the scalability of the method. As more agents are added into the system, credit assignment would become more difficult.
3) Some experimental details need to be clarified further.

**Questions:**

1.	In general, given the complexity of the framework and the dependence on the joint action, I am sceptical about the scalability of the method. It would be interesting to see more results that test the scalability of the method. Convincing results with 8 agents for Cleanup and 1 other environment would significantly increase the strength of the paper.
2.	The observation conversion network is not trained using the ground truth observations of agent j, but instead using the ground truth observations of agent i (this being key to the fully decentralized claim of the work). This makes me wonder why such a network is needed at all? What happens if a network that directly maps from agent i’s observation to agent j’s policy is learned. It would be interesting to see this in an ablation study. In other words, why is the "perspective taking" module not a single network?
3.	How is the SR policy network trained? I am confused why it is trained using RL and not simply using supervised learning using the actual observed actions of agent j.
4.	I am not convinced by the argument that removing the cleaning beam in Cleanup makes the environment harder. Removing the cleaning beam also reduces the dimensionality of the action space. What was the reason behind removing it?
5.	Figure 1 is slightly misleading as it shows separate parameters for the observation conversion network and SR policy network. However, equation 4 shows them having the same parameters.
6.	I am unsure why the policy has a manual epsilon greedy added to it. Actor critic methods are on-policy, the epsilon greedy changes that and I am not sure this manual tuning for the policy is standard. Why was this required?

**Limitations:**

Sufficient details provided.

---

> ### Author Rebuttal · Authors · 2024-08-07
>
> > The complexity of the framework and the scalability of the method.
>
> Thank you very much for your suggestions on LASE’s scalability! To test LASE’s scalability, we have extended Cleanup and Snowdrift as follows:
>
> |  | Map size | Player num | Obs size | Init Waste/ Snowdrift num | Episode length |
> | --- | --- | --- | --- | --- | --- |
> | Cleanup.Extn | 8→12 | 4→8 | 5→7 | 8→16 | 100→150 |
> | Snowdrift.Extn | 8→12 | 4→8 | 5→7 | 6→12 | 50→70 |
>
> Here are the experimental results:
>
> |  | LASE | IA | LIO | SI | A2C |
> | --- | --- | --- | --- | --- | --- |
> | Cleanup.Extn | **56.513** | 20.798 | 1.294 | 3.548 | 0.135 |
> | Snowdrift.Extn | **232.564** | 227.762 | 20.317 | 207.461 | 134.964 |
>
> As shown above, LASE still outperforms the baselines in extended environments. Thus, we say that LASE has a certain degree of scalability. We believe that learning two additional networks for opponent modeling and social relationship inference in a decentralized manner is not overly complicated, ensuring the method's scalability. However, it is undeniable that the current version of LASE requires inferring the social relationships of each agent individually, leading to increased computational complexity as the number of agents increases greatly. The scalability of LASE will be a main focus of our future study. A possible approach is updating the relationship between agents less frequently.
>
> > The explanation of the observational conversion network and the SR policy network. An ablation study about mapping $i$’s observation to $j$’s policy directly.
>
> Thank you very much for your suggestion to replace the PT module with a single network for the ablation study! We trained the network $p(\mathbf{\hat{a_t}} | o^i_t, \mathbf{a_{t-1}})$ with supervised learning by minimizing the MSE loss of the predicted joint action $\mathbf{\hat{a_t}}$ and the real action $\mathbf{a_t}$, and get the following results:
>
> |  | SSH | SSG | Cleanup | Coingame |
> | --- | --- | --- | --- | --- |
> | LASE w/o PT | 18.442 | 118.616 | 37.174 | 29.541 |
> | LASE | 18.948 | 117.784 | 38.736 | 33.467 |
>
> The results show that the two methods perform comparably in SSH, SSG, and Cleanup, while LASE performs better in Coingame.  However, since both approaches predict others' actions based on the same local observations, they essentially serve as opponent modeling tools and thus exhibit similar performance.
>
> We use the PT module in LASE for two main reasons:
>
> **First**, since the SR value network $\phi$ is required to calculate the egocentric $Q$-value when carrying out counterfactual reasoning, and the SR policy network $\mu$ predicts others' actions with the egocentric policy model, both networks share the CNN and some FC layers to extract the observation features.  And they are optimized with the same reward signal under the actor-critic framework (Eq. 5). This sharing is common in actor-critic framework and helps improve training efficiency. Therefore, although the PT module in Figure 1 consists of two networks, the additional parameters and computational overhead mostly stem from the observation conversion network, and it introduces essentially the same amount of parameters as using a single perspective taking network.
>
> **Second**, based on the psychological theory that perspective-taking is a crucial component of cognitive empathy [1], we employ the PT module instead of an end-to-end trained neural network to predict opponent actions, which helps computationally and comprehensively model empathy, the essential mechanism in human society. Since this approach does not result in significant performance loss or increased memory and computational demands, we believe our work contributes new insights into opponent modeling and encourages the community to incorporate human cognitive processes into AI agent design.
>
> [1] Davis, Mark H. "Measuring individual differences in empathy: Evidence for a multidimensional approach." *Journal of personality and social psychology* 44.1 (1983): 113.
>
> > Removing the cleaning beam in Cleanup.
>
> In our implementation, the cleaning beam is replaced by a cleaning action which takes effect only at the waste location, thereby maintaining the dimensionality of the action space. Since the river where the waste accumulates and the apple orchard are located on opposite sides of the map, removing the cleaning beam prevents the agent from directly cleaning waste near the apple orchard. Meanwhile, the time cost of traveling between the river and the apple orchard requires agents to collectively learn a more explicit division of labor strategy in the dilemma. This is why we claim that removal will make the environment harder.
>
> > The parameters of the observation conversion network and SR policy network.
>
> Sorry for the confusion! The observation conversion network $\eta$ and the SR policy network $\mu$ do employ separate sets of parameters. In Equation 4, even though the loss is computed forward through $\mu$, is not used to update $\mu$, but only to update $\eta$. The update method of $\mu$ is in Eq.5.
>
> > Epsilon greedy
>
> We use epsilon greedy mainly to enhance exploration and avoid falling into local optima, which is more likely to occur in mixed-motive games. The introduction of epsilon greedy allows for a controlled level of exploration. The utilization of epsilon greedy is not an anomaly. The classic on-policy algorithm SARSA [2] uses epsilon greedy. A low epsilon value does not cause drastic policy changes, so on-policy methods remain effective. This strategic balance between exploitation and exploration has proven to be empirically sound, fostering a robust approach to algorithmic performance. A similar example is PPO, which uses trajectories sampled from a slightly offset policy for update, but it is still classified as an on-policy algorithm, and it has demonstrated strong empirical performance.
>
> [2] Sutton, Richard S. "Generalization in reinforcement learning: Successful examples using sparse coarse coding." NeurIPS 1995

---

> > ### Comment · Reviewer_AnKj · 2024-08-10
> > **Reply to rebuttal**
> >
> > I thank the authors for the rebuttal. My doubts have been clarified and I have raised my score to reflect the same.

---

> > > ### Author Response · Authors · 2024-08-11
> > > **Thanks for the reply**
> > >
> > > Thank you for your reply! We greatly appreciate the time and effort you put into evaluating our work. Your feedback has provided us with meaningful insights, and we will make sure to address your comments and incorporate the necessary changes in our future revisions.

---

### Official Review · Reviewer_7rD5 · 2024-07-14

**Soundness:** 3
**Presentation:** 3
**Contribution:** 3
**Rating:** 6
**Confidence:** 5

**Summary:**

The authors propose a novel algorithm, LASE, that employs a gifting mechanism in order to steer agents toward equilibria of high social welfare in mixed motive games. A novelty in LASE is that it estimates the "social relationship" between a player and its co-players through a counterfactual Q-value baseline. The authors show that LASE performs favourably across a number of temporally-extended social dilemmas.

**Strengths:**

The paper is well-written and well-presented.

The authors address a gap in the literature which does so far not attempt to estimate the influence of co-player policies on the joint Q-value function. In general, estimating counterfactuals involving Q-values has in the past been found to suffer from high variance issues (such as in [1]). Hence, I find it positively surprising that the authors' method performs decently across several different environments.

[1] Counterfactual Multi-Agent Policy Gradients, Foerster et al., AAAI 2018

**Weaknesses:**

I believe the main weakness of the authors' approach is that, under partial observability, a player cannot generally see all parts of its co-players observations, hence making it impossible to fully reconstruct their policy inputs (hence the restriction to common knowledge fields of view in [2]). I believe the author's algorithm should crucially estimate the uncertainty of their social relationship estimates such as to avoid misunderstandings that could erode trust in real-world situations.

Additionally, I am unsure about the authors' use of the term "empathy" - "theory of mind" would certainly work here, but "empathy" seems like an inherently emotional concept.

[2] Multi-Agent Common Knowledge Reinforcement Learning, Schroeder de Witt et al., NeurIPS 2019

**Questions:**

What do you think could be a real-world application of this line of work?

**Limitations:**

I believe the authors are addressing limitations adequately.

---

> ### Author Rebuttal · Authors · 2024-08-07
>
> > Estimate the uncertainty of their social relationship.
> >
>
> We select the $w^{ij}$ data from the last $10^6$ timesteps of training to calculate their mean value $\overline{w}$ and standard deviation $s$, which estimates the uncertainty of social relationships. The calculation method is as follows:
>
> $$
> \overline{w}^{ij}=\frac{\sum_{t=T_{max}-10^6}^{T_{max}}w_t^{ij}}{10^6}, \overline{w}=\sum_{i=1}^n\sum_{j=1, j\neq i}^n \overline{w}^{ij}
> $$
>
> $$
> s=\frac{\sum_{i=1}^n\sum_{j=1, j\neq i}^n\sqrt{\frac{\sum_{t=T_{max}-10^6}^{T_{max}}(w_t^{ij}-\overline{w}^{ij})^2}{10^6-1}}}{n\times (n-1)}
> $$
>
> We conduct a comparative experiment to replace the input of SR policy network $\hat{o}^j$ with $j$’s real observation $o^j$ . Here is the results:
>
> | $\overline{w}$ | SSH | SSG | Coingame | Cleanup |
> | --- | --- | --- | --- | --- |
> | LASE w/o $o^j$ | $0.07184\pm{0.01921}$ | $0.02341\pm{0.00939}$ | $0.22243\pm{0.18594}$ | $0.34572\pm{0.01509}$ |
> | LASE w/ $o^j$ | $0.06278\pm{0.01136}$ | $0.03157\pm{0.00527}$ | $0.19317\pm{0.05493}$ | $0.29465\pm{0.00889}$ |
>
> The results show that the mean value of LASE's inferred social relationships closely matches that of scenarios with actual observations, although partial observability significantly increases uncertainty. Considering that the social relationships between people in real life tend to be relatively stable and do not change drastically, we think that a possible solution to handle the uncertainty of social relationships is to introduce some smoothing techniques for $w^{ij}$ to reduce the variance of social relationships over time. This approach will be explored in our future work.
>
> Meanwhile, It is important to note that Figure 8 and the corresponding analysis show that LASE is able to correctly infer the relationships with different co-players and respond properly. Specifically, in the experiments conducted in Section 6.3, one LASE interacts with three rule-based co-players: cooperator, defector and random. The results show the $w^{ij}$ given to the cooperative co-player is the largest, significantly higher than that given to the other two co-players. The $w^{ij}$ given to the random co-player comes second, and the smallest is given to the defector. These results demonstrate the consistency between LASE’s estimates of the relations and the ground truth.
>
> Thanks again for your valuable insights! We strongly agree that this issue has an important impact on trust in real-world scenarios, and we will add relevant results and analysis in a revised version of the paper. At the same time, we would also like to thank you for providing the literature about partial observability, which is very helpful to our follow-up work!
>
> > The usage of the term “empathy”
> >
>
> Empathy includes both emotional empathy and cognitive empathy [1]. The former refers to the ability to feel and share others’ emotions. The latter involves imagining and understanding others’ thoughts, feelings and perspectives. In particular, [2] shows that human response is empathically modulated by the learned relations with others. Although there are similarities between Theory of Mind (ToM) and empathy, ToM mainly focuses on attributing mental states to others [1]. Based on [2], we design LASE, incentivizing the cooperative behavior of others through gifts, which is closely related to the emergence of cooperation in real world [3].
>
> [1] De Waal, Frans BM, and Stephanie D. Preston. "Mammalian empathy: behavioral manifestations and neural basis." *Nature Reviews Neuroscience* 18.8 (2017): 498-509.
>
> [2] Singer, Tania, et al. "Empathic neural responses are modulated by the perceived fairness of others." *Nature* 439.7075 (2006): 466-469.
>
> [3] Yalcin, Ӧzge Nilay, and Steve DiPaola. "A computational model of empathy for interactive agents." *Biologically inspired cognitive architectures* 26 (2018): 20-25.
>
> > The real-world application
> >
>
> Our work employs decentralized learning to infer the influence of other agents on oneself in mixed-motive games, enabling broad applicability in complex real-world multi-agent interactions. A potential application scenario is multi-agent automated negotiation. With the rapid advancement and increasing application of machine learning and LLMs, we believe that in the future, agents may assist or even replace humans in various fields such as E-commerce for automated negotiations and autonomous driving.
>
> For example, bargaining in e-commerce is a classic mixed-motive game scenario. In bargaining, both the buyer and seller share the common goal of reaching an agreement. However, each party also aims to maximize their own profit, leading to competition over the transaction price. Consequently, during this process, each side must infer the other's willingness to cooperate and adjust their negotiation strategy accordingly. If the other party shows a high willingness to cooperate, such as when a seller notices that the buyer is very eager to get the good, the seller can propose a higher price. Conversely, if the other side displays a very unfriendly attitude, the agent should consider making some concessions.
>
> Another example is autonomous driving. When two vehicles traveling towards each other meet on a narrow road, they infer the other's manner based on its behavior. If the other one shows courtesy, the focal vehicle proceeds first; if the other one is brash or in a hurry, the focal agent yields to avoid conflict.

---

> > ### Comment · Reviewer_7rD5 · 2024-08-12
> > **Thanks for Your Response, and One More Question.**
> >
> > I thank the authors for their reply. I am satisfied with the authors' response to my concerns about relationship uncertainty and the usage of the term "empathy".
> >
> > Concerning the real-world applicability, I, however, I would like to ask whether and how the authors believe that their work is relevant to LLM agents given the difficulty of performing RL directly with these.

---

> > > ### Author Response · Authors · 2024-08-13
> > > **Two possible approaches to integrating LLM and LASE in negotiation tasks.**
> > >
> > > Given that effective communication and complex strategies in automated negotiation agents often require fluent natural language, it is intuitive to leverage LLMs in this real-world application. Previous work has already formulated the negotiation task and developed datasets that support reinforcement learning, supervised learning, and other methods. [1] Building on this foundation, we propose integrating LLMs into our approach in two ways:
> > >
> > > First, inspired by Cicero [2], which uses RL to train an intent model and generates messages based on that intent using a pre-trained language model, we think that we can prompt an LLM based on the predicted cooperation willingness of the opponent derived from LASE’s SRI module. Considering that our SRI module outputs a scalar value and that LLMs can often misinterpret the magnitude and meaning of numerical values, it may be necessary to establish a mapping system that correlates different cooperation willingness scores with corresponding natural language expressions. For example, if the SRI module predicts that the opponent has a high willingness to cooperate, the LLM, once prompted with this information, might negotiate more assertively, potentially increasing its bargaining position and maximizing profit.
> > >
> > > Second, we can consider transferring the framework of our algorithm to the design of an LLM-based agent, aiming for superior performance in this task. For instance, we could replace the PT module with the LLM's world knowledge by prompting the LLM to infer the opponent's next action based on the current dialogue history. When the LLM observes the opponent's actual action, it could be prompted to engage in counterfactual reasoning, similar to the ReAct [3] framework, updating its belief about the opponent's cooperation willingness. An example template of such a prompt might be: "You initially believed the buyer would offer {}, but the actual offer was {}. Considering your previous belief about the buyer's cooperation willingness was {}, do you think you need to make adjustments? If so, how should it be made?" We hope this approach will help address the issue of LLMs lacking a broader understanding of the overall dialogue progression [4], thereby improving performance in negotiation tasks.
> > >
> > > [1] Post, Thierry, et al. "Deal or no deal? decision making under risk in a large-payoff game show." American Economic Review 98.1 (2008): 38-71.
> > >
> > > [2] Meta Fundamental AI Research Diplomacy Team (FAIR)†, et al. "Human-level play in the game of Diplomacy by combining language models with strategic reasoning." Science 378.6624 (2022): 1067-1074.
> > >
> > > [3] Yao, Shunyu, et al. "React: Synergizing reasoning and acting in language models." arXiv preprint arXiv:2210.03629 (2022).
> > >
> > > [4] Cheng, Yi, et al. "Cooper: Coordinating specialized agents towards a complex dialogue goal." Proceedings of the AAAI Conference on Artificial Intelligence. Vol. 38. No. 16. 2024.

---

> > > > ### Comment · Reviewer_7rD5 · 2024-08-14
> > > > **Thanks for your Response.**
> > > >
> > > > I thank the authors for their response. I appreciate the authors' ideas concerning LLM agents, which I find interesting. I will leave my score unchanged for now.

---

> > > > > ### Author Response · Authors · 2024-08-14
> > > > > **Thanks for your feedback.**
> > > > >
> > > > > Thank you for your continued engagement and valuable feedback on our work! We are grateful for your time and effort in reviewing our paper, and we will carefully consider your questions and comments in the revised version and our future work.

---

### Official Review · Reviewer_vbJc · 2024-07-23

**Soundness:** 4
**Presentation:** 3
**Contribution:** 3
**Rating:** 8
**Confidence:** 4

**Summary:**

This paper introduces LASE (Learning to balance Altruism and Self-interest based on Empathy), a multi-agent reinforcement learning algorithm designed for mixed-motive games. LASE uses a gifting mechanism where agents share a portion of their rewards with others based on inferred social relationships. Counterfactual reasoning determines these relationships by comparing the actual joint action's value to a baseline that averages over the other agents' actions. The authors use a perspective-taking module to predict other agents' policies. Experimental results across various sequential social dilemmas show LASE's ability to promote cooperation while maintaining individual interests. The authors claim the following contributions: (1) a computational model of empathy that modulates responses based on inferred social relationships; (2) a decentralized MARL algorithm that balances altruism and self-interest in mixed-motive games, and (3) theoretical analysis of decision dynamics in iterated matrix games and experimental verification of LASE's performance in sequential social dilemmas.

**Strengths:**

* Novel approach: The introduction of LASE as a mechanism to balance altruism and self-interest in multi-agent settings is innovative and addresses an important challenge in mixed-motive games.

* Theoretical contribution: I think the analysis of the algorithm's behavior in iterated matrix games does a good job of grounding the empirical results in game theory.

* The overall evaluation was sufficient to demonstrate the performance of the algorithm in comparison to well-chosen baselines

**Weaknesses:**

I think the paper is quite strong overall. However, several places can still be improved to increase the potential impact of the work.

* Clarity issues: The paper could benefit from improved clarity in several areas. For example, a bit more clarity on how gifts are determined would be helpful. Consider adding pseudocode for key components like the Social Relationships Inference module or the gifting mechanism and how the modules integrate.

* Insight into counterfactual baseline design choice: The paper's choice of using the average behavior as the counterfactual baseline for determining social relationships warrants further investigation. The authors could consider conducting experiments that vary this baseline, comparing alternatives such as fixed neutral baselines, learned cooperative behavior baselines, or worst-case action baselines. This would provide insights into the robustness of the approach and potentially identify improvements to the algorithm.

* Limited domain coverage: While the paper evaluates LASE across several sequential social dilemmas, which is adequate, the approach could be further strengthened by expanding the range of domains tested. For example, the authors could consider alternatives from MeltingPot domains (https://github.com/google-deepmind/meltingpot).

**Questions:**

Do you have any proposed changes to the paper in response to my critiques?

**Limitations:**

I think there could be more discussion of this, but the current evaluation is sufficient.

---

> ### Author Rebuttal · Authors · 2024-08-07
>
> > Clarify issues
>
> Thanks for your valuable suggestions! The overall framework and flow of LASE as well as the relationship between each module can be seen in Figure 1 in the paper and Algorithm 1 in the appendix. For clarity, here we present the pseudocode for both Social Relationship Inference (SRI) and Gifting modules in detail:
>
> ```python
> # Trajectories collection
> ... ...
>
> # SRI
> for i in Agents:
> 	for j in Agents_without_i:
> 		predicted_obs = obs_conversion(i_obs)
> 		predicted_action_prob = SR_policy(predicted_obs)
> 		for j_virtual_action in all_possible_actions:
> 			SR_q_values.append(SR_value(i_obs, j_virtual_action + others_real_actions))
> 		counterfactual_baseline = predicted_action_prob @ SR_q_values
> 		w_ij = (SR_value(i_obs, real_joint_actions) - counterfactual_baseline) / M # Eq (3)
> 	w_ii = 1 - sum([w_ij for j in Agents_without_i])
>
> # Gifting
> for i in Agents:
> 	i_weighted_reward = 0
> 	for j in Agents:
> 		i_weighted_reward += w_ji * j_reward
>
> # Training with weighted rewards
> ... ...
> ```
>
> These two modules are executed after all agents have completed an entire episode of interaction with the environment, adjusting the rewards to influence subsequent RL training. Specifically, in SRI, each agent $i$ predicts another agent $j$’s action probability using Perspective Taking and assesses the impact of all possible actions of $𝑗$ on its $Q$-value using the SR value network. Then $i$ infers the social relationship $w^{ij}$ with $j$ using Equation 3.
>
> After all agents have completed SRI, the Gifting module is responsible for distributing rewards. Specifically, the weighted reward for each agent $i$ is the sum of the gifted rewards $w^{ij}\times r^j$ from other agents $j$, including $i$’s own reward left to itself $w^{ii}\times r^i$.
>
> > Insight into counterfactual baseline design choice
>
> Thank you very much for your suggestions on designing the counterfactual baseline! We apologize for the confusion we may have encountered while attempting to implement the three different baselines you mentioned. We are not entirely sure if we fully understood your points and would greatly appreciate the opportunity to discuss this matter in more detail to ensure we correctly implement your suggestions!
>
> - Fixed neutral baseline: If you are referring to using a fixed hyperparameter as a neutral baseline, we believe that this approach may not be appropriate due to the continuously evolving $Q$-values throughout the RL learning process. If you are referring to the neutral value of $i$'s $Q$-values conditioned on the various possible actions of $j$, we think that our ablation study **LASE w/o**, where $\sum_{a_t^{j'}}\frac{1}{|\mathcal{A}^j|}Q^i(o^i_t,(a_t^{-j},a_t^{j'}))$ is used as a counterfactual baseline, has already addressed this.
>
> - Learned cooperative behavior baselines: Are you referring to "cooperative behavior" as $\text{max}_{a^{j'}_t} Q^i(o^i_t, (a_t^{-j},a_t^{j'}))$? If so, based on Equation 3, this would lead to $w^{ij} \leq 0$, resulting in no gifting occurring. This would cause the algorithm to fail and revert to the A2C baseline.
> - Worst-case action baselines: We interpret this baseline as $\text{min}_{a^{j'}_t}Q^i(o^i_t, (a_t^{-j},a_t^{j'}))$. We replace LASE's counterfactual baseline with the Worst-case Action Baseline (WAB), and the self-play results are as follows:
>
>     |*Self-play*|SSH|SSG|Coingame|Cleanup|
>     |---|---|---|---|---|
>     |WAB|17.624|118.731|0.009|43.741|
>     |LASE|18.948|117.784|38.736|33.467|
>
>   WAB outperforms LASE on SSG and Cleanup, which is because using $\text{min}_{a^{j'}_t}Q^i(o^i_t, (a_t^{-j},a_t^{j'}))$ as the baseline results in $w^{ij}\geq 0$. This overly optimistic estimation of social relationships promotes group cooperation in self-play settings, but it also raises the risk of exploitation when interacting with unknown agents. To demonstrate this, an adaptive experiment on WAB is conducted in the same manner as described in Section 6.3 of the paper. A WAB is trained with three rule-based agents and three A2C agents, respectively. We record the mean gifting weights assigned by the WAB to other agents and the rewards of the WAB after gifting:
>
>     |*Gifting weight*|Cooperator|Random|Defector|A2C_1|A2C_2|A2C_3|
>     |---|---|---|---|---|---|---|
>     |WAB|0.321|0.254|0.184|0.266|0.221|0.222|
>     |LASE|0.268|0.091|0.027|0.0391|0.1339|0.2194|
>
>     |*Reward after gifting*|Rule based agents|A2C agents|
>     |---|---|---|
>     |WAB|5.778|4.467|
>     |LASE|**15.129**|**8.752**|
>
>     Due to WAB giving excessive gifts to other agents and not receiving reciprocation as it would in self-play, the actual reward obtained by WAB is significantly lower than that of LASE. This demonstrates the weakness of WAB being easily exploited.
>
>     Additionally, we observe that in Coingame, the social relationship converged to $w^{12}\approx w^{21}\approx 1$, indicating that the agents' optimization objective shifted towards maximizing the opponent's rewards. In Coingame, only collecting coins of the opponent's color impacts its reward. Finally, the two agents converged to a behavior of not collecting any coins, demonstrating that excessive gifting is detrimental to learning in this environment.
>
> > Limited domain coverage
>
> Thank you very much for your suggestions! As a standardized testing platform that integrates various environments including cooperation, competition, and mixed motives, MeltingPot can further assist in testing the scalability and robustness of algorithms. We will consider selecting some of these environments as experimental settings in our future work to enhance the persuasiveness of our paper.
>
> Thank you again for your insightful comments! We will make corresponding changes in the revised version of the paper based on your and other reviewers’ suggestions, including but not limited to improving and clarifying the description of the algorithm, adding discussions and experiments on the counterfactual baseline, and expanding the experimental environments.

---

### Decision · Program_Chairs · 2024-09-25

**Decision:**

Accept (poster)

**Comment:**

All reviewers feel the paper makes a sound and useful contribution.  The authors effectively addressed the reviewers' concerns during the discussion phase.  I encourage the authors to implement what they mentioned in the discussion to improve the paper.